# Localized Colonic Small-Cell Carcinoma with Pathological Complete Response after Neoadjuvant Cisplatin and Etoposide: A Case Report

Víctor Alía Navarro [1,*] , Íñigo Martínez Delfrade [1,2], Belén De Frutos González [1,2], Blanca Morón García [1,2], Ana María Barrill Corpa [1], Pilar Sotoca Rubio [1], Beatriz Peñas García [3], Ana Ferrer Gómez [4] , Cristian Perna Monroy [4] and Reyes Ferreiro Monteagudo [1,2,5]

1    Medical Oncology Department, Ramón y Cajal University Hospital, 28034 Madrid, Spain; mariareyes.ferreiro@salud.madrid.org (R.F.M.)
2    Ramón y Cajal Health Research Institute (IRYCIS), 28034 Madrid, Spain
3    Gastroenterology Department, Ramón y Cajal University Hospital, 28034 Madrid, Spain
4    Pathology Department, Ramón y Cajal University Hospital, 28034 Madrid, Spain
5    Biomedical Research Network in Cancer (CIBERONC), 28029 Madrid, Spain
*    Correspondence: victor.alia@salud.madrid.org

**Abstract:** Extrapulmonary small-cell carcinoma (SCC) is a rare neoplasm that shares certain features with its pulmonary counterpart and occurs predominantly in the gastrointestinal tract (GIT). It is a high-grade and poorly differentiated neuroendocrine tumor, usually diagnosed in advanced stages, with a poor prognosis and few therapeutic options in that setting. This is a case report of a 77-year-old Spanish male patient with localized SCC of the colon, who presented a pathological complete response in the surgical specimen after neoadjuvant chemotherapy with cisplatin and etoposide. To date, 5 years after surgery, the patient remains without evidence of tumor recurrence. As clinical guidelines for the management of this entity are lacking, and therefore its management has not been standardized, an attempt to summarize the current evidence in the literature was made.

**Keywords:** small-cell carcinoma (SCC); extrapulmonary small-cell carcinoma; colonic small-cell carcinoma; pathological complete response; neoadjuvant chemotherapy; case report

## 1. Introduction

Small-cell carcinoma (SCC) is a high-grade and poorly differentiated neuroendocrine neoplasm [1]. SCC is categorized as a neuroendocrine carcinoma. In the past, it was thought to originate from the dedifferentiation of Kulchitsky cells, which are pulmonary neuroendocrine cells present alongside the basement membrane of the epithelium of the bronchi and bronchioles [2], and enterochromaffin cells found in the epithelium of the gastrointestinal tract. These cells are characterized by the presence of neurosecretory granules, which explains the usual diffuse reactivity for neuroendocrine markers in immunohistochemical studies. Although its pathogenesis is not fully understood to date, the most widely accepted theory points to its origin from a multipotent stem cell with divergent differentiation potential [3].

Lung is certainly the most common anatomical location of the disease, as it is a typical tumor of heavy smokers. Small-cell lung cancer (SCLC) currently represents around 15% of all new lung tumor diagnoses. This represents an estimated incidence of about 250,000 cases, and more than 200,000 deaths each year worldwide, according to the literature [4]. This neoplasm is characterized by a high proliferation rate and an advanced vascular invasion with a peculiar tendency to spread and generate locoregional lymph node and distant metastases, conferring a poor prognosis and low survival rates despite the interest in ongoing research into new therapies for these patients. Surgery alone remains insufficient in this setting, and multidisciplinary treatment including radiation or

chemotherapy agents is required. Regarding the mutational status of SCLC, a clear genetic imprint of tobacco smoking has been found. Simultaneous inactivation of two tumor suppressor proteins p53 and Rb plays a key role in the biology of this tumor and is present in most cases of SCLC [5].

Extrapulmonary SCC is a rare malignancy that constitutes approximately 2.5–5% of all cases of SCC, with the gastrointestinal and genitourinary tracts being the most common locations [6]. More infrequently, breast [7], head and neck [8–10], uterine cervix [11], ovary [12], prostate [13], kidney [14] and a wide variety of other anatomical locations of SCC cases have been reported. Specifically, the gastrointestinal variant shares several features with its pulmonary counterpart, suggesting a common biological origin. However, gastrointestinal SCC is considered an identity of its own. This is because it presents a higher proportion of local and locoregional disease, a lower correlation with smoking burden and a better overall prognosis according to the latest series published in the literature, justifying the higher survival of these patients compared to its pulmonary equivalent.

The 2019 World Health Organization (WHO) classification of gastrointestinal neuroendocrine tumors categorize this neoplasm as a small-cell neuroendocrine carcinoma (SCNEC). Multifocality is a common finding in this malignancy, and these tumors are categorized as mixed neuroendocrine-non-neuroendocrine tumors (MiNENs) [15].

Within gastrointestinal SCC, esophagus is the most common location in around 50–55% of cases according to the largest series studied [16]. Colonic localization represents around 13% of gastrointestinal cases, which is a very rare entity with few cases reported worldwide to date.

Regarding colon SCC, it is a rare neoplasm that constitutes around 0.2–0.8% of all colonic tumors [17]. It occurs predominantly in patients aged over 50 years old. A slight male predominance is observed, and the vast majority of tumors are located in the cecal and sigmoid regions. Thus, this anatomical distribution is similar to that of the primary colonic adenocarcinoma. It frequently arises associated in an adenomatous polyp and mixed with other histologic types [18].

Sometimes, these tumors are discovered in the background of inflammatory bowel diseases, suggesting a higher incidence in this group of patients [19,20]. Additionally, there are some reported cases of SCCs arising in colon duplication cysts [21].

Colon SCCs are clinically aggressive tumors and, unfortunately, most patients present with overt distant metastases at the time of diagnosis. The most frequent sites of distant metastasis are lymph nodes (45%), liver (40%), lung (10%) and bone (5%) [22]. Therefore, the mean overall survival (OS) in this context is approximately 6 months, and 1-year survival rate is only around 15% [17,18].

The case report presented below is of great interest because it involves a patient with an incidental diagnosis of localized primary SCC in the left colon, who presented a pathological complete response in the surgical specimen after four cycles of neoadjuvant chemotherapy with cisplatin and etoposide, remaining free of recurrence to date. There is currently little published literature on localized colonic SCC due to its rarity, and no other cases of complete pathological response after exclusive neoadjuvant chemotherapy have been reported, to our knowledge.

## 2. Case Presentation

A 77-year-old Spanish man had a history of well-controlled arterial hypertension, dyslipidemia, and obstructive sleep apnea–hypopnea syndrome in treatment with nocturnal continuous positive airway pressure. He was a heavy ex-smoker with a cumulative pack-year index of 96, and he had no history of significant alcohol or other drugs significant intake. There was no known history of environmental exposure to asbestos or other occupational carcinogens. He had no history of familial aggregation of cancer.

Three years prior, he underwent sigmoidectomy due to a localized sigma adenocarcinoma pT2N0, incidentally diagnosed on a screening colonoscopy. The surgical specimen showed tumor-free margins, so adjuvant treatment was not indicated. Therefore, a

subsequent follow-up with periodic colonoscopy and computed tomography (CT) was carried out.

In a subsequent medical check in the Medical Oncology consultation, the patient was asymptomatic, with an Eastern Cooperative Oncology Group (ECOG) score of zero and a Karnofsky Performance Status (KPS) of 90%. A thoracoabdominal CT scan, performed as follow-up of the previous tumor, revealed a 21 × 8 mm lung lesion in the left upper lobe of indeterminate nature. No suspicious lesions were identified in the colorectal tract or other locations in this imaging test.

After this finding, a fibrobronchoscopy with biopsy was performed. The pathology showed no evidence of malignancy, but instead revealed nonspecific inflammatory changes. In addition, a positron emission tomography (PET) scan with 18-fluorodeoxyglucose was performed, with no increased metabolism in the pulmonary nodule. Nevertheless, a significant increase in glycidic metabolism was observed in a small lesion in the left colon, with a standardized uptake value (SUV) of 4.75.

Blood tumor biomarkers performed at that time were all within the range of normal levels (prostate-specific antigen 2.3 ng/mL, enolase 21.4 ng/mL, chromogranin A 40.1 ng/mL, carcinoembryonic antigen 1.7 ng/mL and CA19-9 antigen 2.4 U/mL).

A colonoscopy was performed due to the increased metabolism observed at that level in the PET. It revealed, in the left colon, 35 cm from the anal margin, a sessile mamelonated lesion of about 2 cm in diameters with a dark pigmented appearance, friable and surrounded by an erythematous mucosa with a macroscopic inflammatory appearance (Figure 1). Biopsies were taken and ink tattooing was performed.

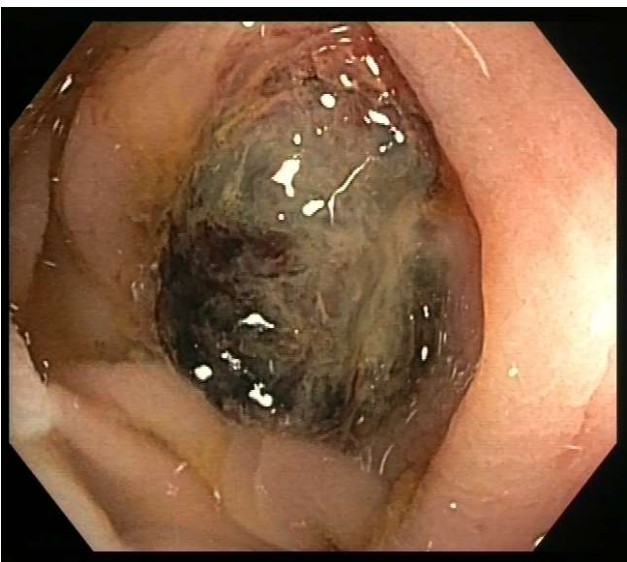

**Figure 1.** Colonoscopy image of the sessile tumor lesion.

A histological study showed an undifferentiated high-grade small-cell carcinoma with extensive areas of necrosis and chromatin stretching. The immunohistochemical study showed paranuclear positivity for cytokeratin (CK) AE1/AE3 and negativity for CD45, CK7, CK20 and TTF1. Regarding neuroendocrine markers, the tumor presented synaptophysin positivity, but chromogranin and S100 negativity and a ki67 proliferation index above 90% (Figure 2).

A diagnosis of colon primary SCC was asssigned. Subsequently, due to the unusual presentation, the case was discussed in the Tumor Committee with a multidisciplinary team made up of oncologists, radiotherapists, pathologists, radiologists, nuclear medicine physicians, digestive endoscopists and general surgeons. Regarding the pulmonary nodule, it disappeared in a follow-up CT performed after a course of antibiotic therapy due to underlying catarrhal symptoms. Therefore, it was concluded that the etiology of the prior lung lesion was infectious and not tumoral. It was decided to perform a brain nuclear

magnetic resonance (NMR) due to the known neurotropism of this tumor histology in its pulmonary form as SCLC. The brain NMR ruled out secondary metastatic involvement at that level.

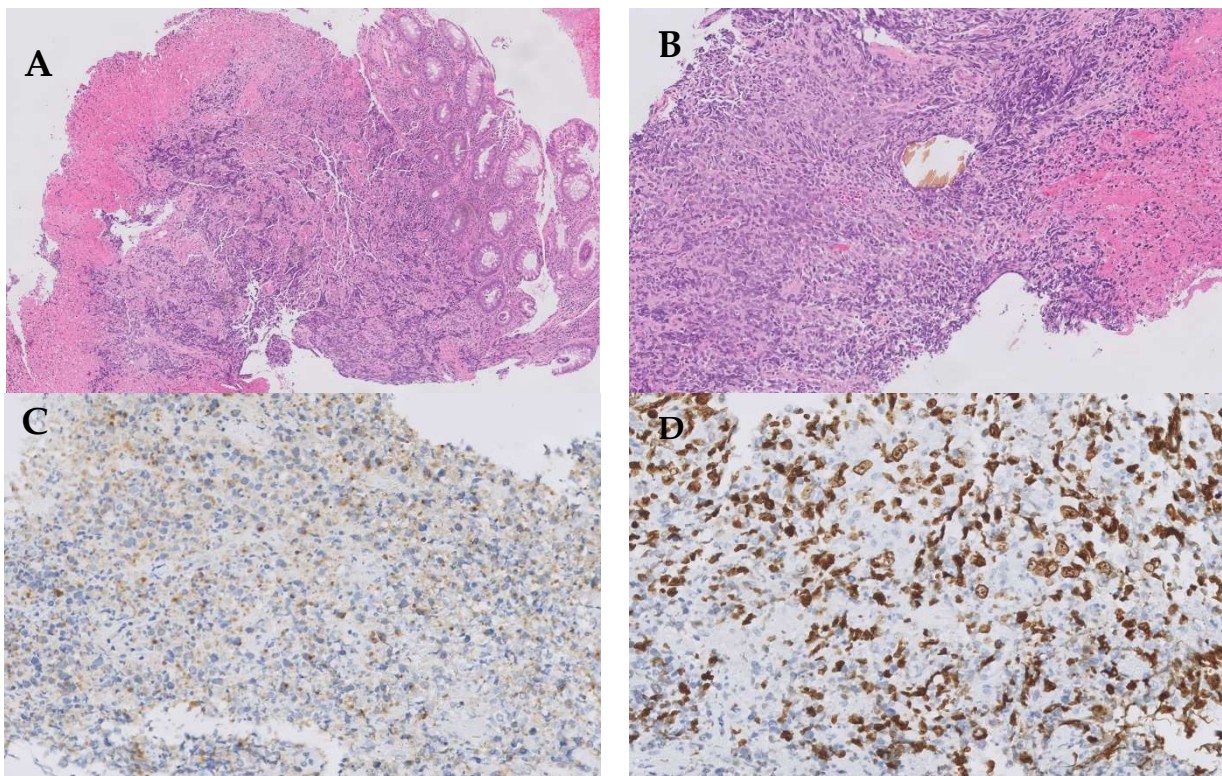

**Figure 2.** Light microscopy image of tumor biopsy. (**A**) Hematoxylin–eosin (H&E) stain 5×. Large intestine mucosa infiltrated by a solid neoplasm composed of small cells with abundant necrosis. (**B**) H&E stain 10×. Tumoral cells show scant cytoplasm and nuclear pleomorphism. (**C**) Tumoral cells stain positive for synaptophysine. (**D**) ki67 proliferation index above 90%.

Once localized colon primary SCC was confirmed, it was decided to start treatment with platinum and Etoposide scheme with neoadjuvant intention. The usual doses for the treatment of pulmonary SCC were used based on previous evidence. At that time, the patient did not exhibit any ineligibility criteria for treatment with platinum, so the regimen consisted of Cisplatin 80 mg/m$^2$ given intravenously on Day 1 and Etoposide 100 mg/m$^2$ intravenously on Days 1, 2, and 3, repeated every 21 days. After four well-tolerated cycles of treatment, a total PET-CT post-neoadjuvant chemotherapy was performed for reassessment, showing a metabolic complete response of the left colonic lesion.

Laparoscopic left hemicolectomy was performed 4 weeks after the completion of the last cycle of chemotherapy, without remarkable postoperative complications. The pathology showed no evidence of malignancy in the surgical specimen, which included the tattoo and the previous biopsy bed. One lymph node was isolated without evidence of malignancy.

The multidisciplinary committee decided to rule out prophylactic cranial irradiation (PCI) due to the expected absence of benefit after a review of the literature [23].

Subsequently, the patient started check-ups with periodic colonoscopies and total-body CTs, initially every 3 months for the first 2 years and later every 6–12 months, with no evidence of tumor recurrence and remaining asymptomatic to date, 5 years after the surgery.

## 3. Discussion

This case report shows a pathological complete tumor response in a patient with a localized SCC of the colon, achieved after neoadjuvant chemotherapy, with a subsequent long-term survival after surgery, and with no evidence of tumor recurrence to date.

Regarding the diagnostic process, colorectal SCCs are morphologically identical to SCLCs. These small basophilic cells usually present densely packed and granular nuclear chromatin, with minimal amounts of cytoplasm. In fact, the size of their nuclei is approximately twice the size of a lymphocyte [24].

The expression of neuroendocrine markers is usual but inconsistent in these neoplasms. Therefore, some experts in this area suggest that pathological diagnosis of these tumors should rest on routine haematoxylin–eosin stains, based on cellular morphology. The presence of expression of neuroendocrine markers offers strong support to the diagnosis of these malignancies, but their absence should not completely rule it out if diagnostic suspicion is high [25]. In our case report, immunophenotype revealed the presence of diffuse positivity for synaptophysin, but negativity for chromogranin. However, the morphology of the cells and the negativity of the rest of the markers for alternative tumors allowed us the assignment of the diagnosis. It has been reported that around 70% of gastrointestinal SCCs aberrantly overexpress Human Achaete-Scute Homologue Gene-1 Protein (hASH1). Gastrointestinal neuroendocrine cells, carcinoids and adenocarcinoma seldom express hASH1, so it has a better sensibility and specificity than other classic neuroendocrine markers, and it may be useful as a new biomarker for gastrointestinal SCCs [26]. New second-generation neuroendocrine immunohistochemical markers with greater specificity are being explored. Some of them are ISL LIM Homebox 1 (ISL1), INSM Transcription Repressor 1 (INSM1), or Secretagogin (SECG), which could be welcomed into routine clinical practice in the coming years [27].

In this context, this is an unusual case report since most of these extrapulmonary SCC cases are usually diagnosed in advanced stages. This is due to its known metastatic potential, with a rapid and aggressive clinical course, a high risk for recurrence, and therefore poor survival rates. As expected, patients with localized stages have a better outcome. Although there is little literature in this setting, a median survival rate of 21.9 months was observed in patients with gastrointestinal SCC with limited disease compared with those advanced stages with a median survival rate of 5.8 months [17]. The largest cohort of patients with colon SCC to date confirms its aggressiveness. Most cases of this cohort presented advanced disease at the time of diagnosis, with poor prognosis, but a benefit of chemotherapy in terms of progression-free survival and overall survival was observed. However, there was no clear benefit of surgery in this scenario [28].

Despite the fact that this entity has usually been treated similarly to pulmonary SCC, currently, there are no clinical guidelines or consensus to standardize its management in clinical practice due to its infrequency.

A study based on cases from Surveillance, Epidemiology and End Results (SEER) aimed to determine the benefit of each treatment modality in pulmonary and extrapulmonary SSC. A similar survival was found in both groups, with a 5-year survival of 7% and 5%, respectively. Additionally, a potential role of locoregional therapy with radiation and surgery in those extrapulmonary cases was observed [29,30].

Some retrospective studies aimed to assess the role of surgery, local radiation and systemic therapies in the treatment of these patients. At this time, there is evidence enough to justify a different approach and management of the localized and the advanced disease.

Localized disease, as shown in this case report, is potentially curable with surgery and/or other locoregional therapies. Due to the known chemosensitivity of this tumor, perioperative strategies such as neoadjuvant and/or adjuvant chemotherapy may allow better control of the disease, and therefore an increase in the success of the surgery. Combined chemoradiotherapy appears to be an alternative to surgery, as an occasional pathological complete response has been observed in several case reports [31,32].

Meanwhile, advanced disease has a worse prognosis, so palliative treatment is indicated with systemic chemotherapy, with schemes usually administered in pulmonary SCC, based mainly on platinum agents and Etoposide. In those chemotherapy ineligible patients, due to underlying frailty or a poor performance status, the best supportive care is indicated.

There is a low incidence of brain metastasis in gastrointestinal and genitourinary SCC in the literature, so PCI is not routinely recommended, even in the metastatic setting. However, the frequency of brain metastasis in prostate and head and neck SCC is much higher, so PCI should be considered for these patients [23].

Regarding immunotherapy agents, it has recently become part of the first line of standard treatment in extensive SCLC [33]. Likewise, there are other ongoing studies testing new bispecific antibodies in patients with recurrent SCLC, with promising preliminary results [34]. There was a small phase II basket clinical trial testing dual anti-CTLA4 (Ipilimumab) and anti-PD-1 (Nivolumab) blockade every 2 weeks until disease progression or unacceptable toxicity. It included a cohort of high-grade extrapulmonary neuroendocrine neoplasms. This cohort included six patients with primary SCC gastrointestinal tumors (two patients with SCC in the gastroesophageal junction, two in the pancreas, one in the cecum and one in the rectum), showing modest results. Among those six patients, four had confirmed partial response. The overall 6-month progression-free survival (PFS) rate was 32% (16–61%) with a median PFS of 2.0 months and a median OS of 8.9 months. A total of 37% of patients presented Grade 3/4 immune-related adverse events [35].

As previously mentioned, colon SCC has a high risk for recurrence, so a close radiological follow-up after treatment is recommended. The best follow-up strategy has not been established in this setting. Based on previous reports, it is common to perform whole-body screening with CT, magnetic resonance imaging or PET-CT every 3 months initially, including brain assessment [36].

In summary, more research efforts are needed to improve survival outcomes and quality of life in patients with gastrointestinal SCC. It is a priority to design and carry out prospective studies and clinical trials to provide greater evidence in the management of this malignancy in order to assess the best strategy for treatment. To date, the limited evidence available comes from few case reports and small retrospective studies because of the uncommonness of these neoplasms. Nevertheless, as previously stated, there is increasing evidence that the extrapulmonary SCC has a different behavior compared to its pulmonary counterpart. This fact shows that extrapulmonary SCC is probably a different entity with its own features, and therefore it may require a different approach.

**Author Contributions:** Conceptualization, V.A.N., Í.M.D., B.D.F.G., B.M.G., A.M.B.C., P.S.R., B.P.G., A.F.G., C.P.M. and R.F.M.; Writing—original draft preparation, V.A.N.; Writing—review and editing, V.A.N. and R.F.M.; Supervision, V.A.N. and R.F.M. All authors have read and agreed to the published version of the manuscript.

**Funding:** This research received funding from Fundación para la Investigación Biomédica del Hospital Universitario Ramón y Cajal (FIBio-HRC).

**Institutional Review Board Statement:** Not applicable.

**Informed Consent Statement:** Written informed consent has been obtained from the patient to publish this paper.

**Data Availability Statement:** The data presented in this study are available in this article.

**Conflicts of Interest:** The authors declare no conflict of interest.

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
