# Peer review of "Localized Colonic Small-Cell Carcinoma with Pathological Complete Response after Neoadjuvant Cisplatin and Etoposide: A Case Report"

_curroncol, doi:10.3390/curroncol30090613_

Round 1
Reviewer 1 Report
The authors have created a case report of a localized small cell GI carcinoma who had a CR on neoadjuvant chemotherapy. I feel this was a well thought out and organized case report. It absolutely adds to the literature and I feel is important to be published as it provides a potential treatment recommendation for an extremely rare tumor case.
With that being said, I did have a couple of comments and questions.
1. As this case is illustrating the importance of neoadjuvant treatment in this patient, I think it would be important to provide the dose and scheduling for the cisplatin and etoposide. I imagine if an oncologist is looking to this case report for ideas how to treat a similar patient in their practice, the information about the regimen would assist the reader.
2. With regards to the colonoscopy that was done, I assume it was done for screening purposes? This was not mentioned so I would suggest adding why it was done.
3. the 2cm lesion that was identified on colonscopy, it is not mentioned if it was seen on the CT. I would suggest highlighting that information as well.
4. for the tumor markers, it was mentioned they were all negative. I would suggest actually putting in the actual numbers. Negative could mean they were all zero or they were in the normal range.
5. In the introduction, there seems to be some missing references/citations. "SCC is a high-grade, poorly diff...", and paragraph 2 of the intro should have references associated with them.
6. It would be nice to have a table of all of the GI small cell case report/series looking at location, treatments and outcomes. This would make this case report a one-stop shop for all information for these kinds of rare malignancies.
There are some spelling and grammatical errors that will need to be addressed. For example remais in the abstract I assume should be remains. Genitourinay in the introduction I assume should be genitourinary.
There are some sentence structure issues as well that will need to be fixed. for example in the case presentation, "Ex-smoker with cumulative pack-year index of 96, non drinker." is not a complete sentence.
Author Response
Dear reviewer,
I greatly appreciate your thorough and detailed review of our manuscript.
Here I attach the revised version of the paper with the changes you suggested, and the spelling and grammatical corrections that I have detected.
Kind regards,
Víctor Alía.

Reviewer 2 Report
SCC is indeed a rare disease, and experience in clinical management is still very limited. In order to improve the report of this article, it is suggested that:
1 This article mentions CR after treatment, and more images are needed as a basis, such as providing comparable PET examinations, comparable colonoscopy images, and MR or CT to indicate the status of CR.
2 It is mentioned in the original article that PET is also suspected to be a metastatic lesion originating in the lung, so what is the clinical significance of this metastatic CR? What is the relationship between whole body treatment and him? How to explain the relationship between the two?
3 Need to explain why additional LHC is needed if CR is considered to be reached?
N/A
Author Response
Dear reviewer,
I greatly appreciate your review of our manuscript.
The CR refers to a pathological response (the lesion in the colon was not observed in CT, but only as a hypermetabolism in PET. Therefore, no control imaging test was performed after neoadjuvant treatment prior to surgery). I have modified the manuscript to clarify this issue.
For this reason, right hemicolectomy was performed after neoadjuvant therapy due to the difficulty in assessing a radiological response.
Regarding the lung lesion, it was observed in the CT, showing absence of hypermetabolism in PET. Due to the presence of cold symptoms at the time of the CT scan, a cycle of antibiotic was performed, and in a follow-up CT (prior to the start of neoadjuvant therapy), there was no evidence of said lesion. Therefore, it was concluded that the etiology of the lung lesion was infectious, and not a metastatic lesion.
Here I attach the revised version of the paper with some changes trying to clarify some parts of the case, and the spelling and grammatical corrections of the mistakes I have detected.
Kind regards,
Víctor Alía.

Reviewer 3 Report
The authors persent a Clinical Case of a Small Cell Carcinoma localized in the left Colon which deloped pathological complete response after neoadjuvant chemotherapy.
The manuscript is well written. The clinical case has been well studied and it is well persented in the paper.
The interest of the paper lies on the rarity of this entity. The number of small cell carcinomas of the large intestine that have been published is very reduced. The authors accompanies the description of the clinical case with a brief review of the literature which is very informative and that justifies the publication of the clinical case
The english language is correct and it need only minor corrections.
Author Response
Dear reviewer,
I greatly appreciate your thorough and detailed review of our manuscript.
Here I attach the revised version of the paper with some changes suggested, and the spelling and grammatical corrections that I have detected.
Kind regards,
Víctor Alía.
